# LOSSLESS DATA COMPRESSION WITH TRANSFORMER

## ABSTRACT

Transformers have replaced long-short term memory and other recurrent neural networks variants in sequence modeling. It achieves state-of-the-art performance on a wide range of tasks related to natural language processing, including language modeling, machine translation, and sentence representation. Lossless compression is another problem that can benefit from better sequence models. It is closely related to the problem of online learning of language models. But, despite this ressemblance, it is an area where purely neural network based methods have not yet reached the compression ratio of state-of-the-art algorithms. In this paper, we propose a Transformer based lossless compression method that match the best compression ratio for text. Our approach is purely based on neural networks and does not rely on hand-crafted features as other lossless compression algorithms. We also provide a thorough study of the impact of the different components of the Transformer and its training on the compression ratio.

## 1 INTRODUCTION

Lossless compression is a class of compression algorithms that allows for the perfect reconstruction of the original data. In the last decades, statistical methods for lossless compression have been dominated by PAQ-type approaches (Mahoney, 2005). The structure of these approaches is similar to the Prediction by Partial Matching (PPM) of Cleary & Witten (1984) and are composed of two separated parts: a predictor and an entropy encoding. Entropy coding scheme like arithmetic coding (Rissanen & Langdon, 1979) are optimal and most of the compression gains are coming from improving the predictor. The predictor estimates the probability of a token given its past, which is a standard sequence modeling problem. In most PAQ-type approaches, this predictor relies heavily on hand-crafted features, but recent work has used the close connection between sequence modeling and machine learning to add neural networks based methods in the predictor. In particular, CMIX (Knoll, 2014) has successfully added a Long-Short Term Memory (LSTM, Hochreiter & Schmidhuber (1997)) to the predictor of PAQ8. However, these approaches are still heavily relying on hand-crafted features along with the neural networks, and purely neural network based models are still far from the state of the art (Bellard, 2019).

In this work, we aim at building a purely neural network based model that compete the state of the art. We follow Knoll & de Freitas (2011) and formulate the token prediction problem solved by the predictor as the online learning of a sequence model. As opposed to previous work, we propose to use a Transformer (Vaswani et al., 2017a) to replace entirely the predictor of PAQ8 and its handcrafted features. Transformer has emerged as the standard model for sequence modeling in natural language processing, achieving state-of-the-art performance in numerous applications, including, but not limited to, language modeling (Dai et al., 2019; Sukhbaatar et al., 2019b), machine translation (Vaswani et al., 2017a; Ott et al., 2018), or sentence representation (Devlin et al., 2018; Yang et al., 2019). As opposed to the recurrent neural networks (RNNs, Elman (1990)) used in lossless compression, they are able to capture very long-term dependencies by allowing direct connections between long-distance symbols.

However, the training of a Transformer is unstable and slow, limiting its application to the online setting of lossless compression. We propose several improvements to its architecture and training to accelerate and stabilize its online training. We show that our neural network approach matches the state of the art on the competitive benchmark `enwik8` used in the Hutter prize (Mahoney, 2005). We also provide a thorough study of all the important components of a Transformer and its training to achieve this performance.

## 2 RELATED WORK

In this section, we briefly introduce statistical based methods for lossless compression, and focus on PAQ based compression methods as well as purely neural network based methods.

**Statistical based methods.** A popular statistical based method for lossless data compression is the Prediction by Partial Matching (PPM) of Cleary & Witten (1984). It is based on a predictor and a entropy coding, that often takes the form of an arthimetic coding (Rissanen & Langdon, 1979; Witten et al., 1987b), but could also be a Huffman coding (Huffman, 1952b). There are many variants of PPM such as PPM*C (Cleary & Teahan, 1997) or cPPMI-64 Shkarin (2002). Another statistical based method is the stochastic sequence memoizer of Gasthaus et al. (2010) that is based on a nonparametric Bayesian model. In particular, the stochastic memoizer of Wood et al. (2009) shares many similarities with PPM and reaches similar compression performance. However, most of these models are now surpassed by PAQ based methods like CMIX (Knoll, 2014).

**PAQ based compression methods.** The series of PAQ algorithms, including PAQ8 (Mahoney, 2005), are statistical based methods designed to operate at bit-level, making it applicable to any type of data (Salomon & Motta, 2010). There are also different version of PAQ8 specialised in different data types for better compression performance. PAQ uses a context mixer that follows the same structure as PPM. However, unlike PPM, the predictor of the context mixers combines the probability of different prediction models (Knoll & de Freitas, 2011). More recently, a PAQ based model called CMIX (Knoll, 2014) has been the state of the art for lossless text compression in terms of compression rate. As opposed to PAQ8, CMIX uses a Long Short Term Memory network (LSTM, Hochreiter & Schmidhuber (1997)) in their probabilistic model. The context mixer used in CMIX is also based on a gated linear network (Veness et al., 2017).

**Neural Network based compression method.** More recently, there has been some attempts to use purely neural network based approaches for the probabilistic model. Notably Knoll released lstm-compress (Knoll, 2015), which uses only the LSTM introduced in CMIX and the preprocessing of CMIX. Bellard (2019) investigates the performance of pure neural approaches with modern and large neural networks such as LSTM and Transformer. As well as our approach, all of these approaches use arithmetic coding as entropy coding scheme.

## 3 LOSSLESS DATA COMPRESSION

In this section, we briefly introduce the core components of statistical methods for lossless compression. As opposed to lossy compression where it is allowed to lose information about the orginal data in the reconstruction, lossless compression methods restore the data to its original value.

Lossless compression methods are composed of two main elements: a predictor and an entropy coding scheme. They are used successively to encode an input stream of tokens $s_0, \ldots, s_T$. More precisely, for each token, the predictor first evaluates its probability given its past, then the entropy coding scheme uses this probability to store the token into a lossless code. More precisely, if we assume that $t$ tokens $s_0, \ldots, s_{t-1}$ are already compressed into a lossless code, the predictor computes the probability of the next token to be equal to its value $s_t$ given all the preceding tokens, i.e., the predictor computes $p(s_t \mid s_{t-1}, \ldots, s_0)$. Then, the entropy coding scheme encodes the token $s_t$ into a lossless code of bit-length $\log_2 p(s_t \mid s_{t-1}, \ldots, s_0)$. Once the token $s_t$ has been encoded, these two steps are repeated on following tokens until the end of the sequence.

The operations performed by the predictor are identical during compression and decompression phases, i.e., in both cases it predicts the next token given all the preceding tokens. The only difference is that, during the compression, the preceding tokens are taken from the input file, while during the decompression, they are generated by the entropy coding scheme. Note that, as opposed to the predictor, the role of the entropy coding scheme differs during the two phases since, during decompression, it acts as a decoder, i.e., given the compressed representation and the sequence of predicted probability distributions $(p(s_t|s_{t-1}, \ldots, s_0))_{0 \leq t \leq T}$ it restores the original sequence of tokens.

There are many efficient and optimal methods for the entropy coding scheme. The source coding theorem Shannon (1948) has introduced the fundamental idea that any probability distribution can be

encoded into a lossless code. For a given data point, the bit-length of this code is equal to the negative log probability estimated by the model. Thus, efficient probabilistic models are crucial to design effective compression scheme since the smallest codes are obtained when the model estimates the true data distribution. Depending on the type of probabilistic model considered, several compression schemes have been developed to implement this idea in practice, namely Huffman coding Huffman (1952a), arithmetic coding Witten et al. (1987a) and asymmetric coding Duda (2009). The basic idea of all these entropy coding schemes is to assign shorter codes to the more likely tokens. In this work, we use the arithmetic coding scheme.

## 4 OUR APPROACH

In this section, we introduce our approach to lossless data compression using neural networks. First, we briefly review transformer networks, which are at the core of our method. We then discuss modifications to the standard model, to make it more suitable to long sequence modeling. Finally, we propose a strategy to make the online learning of large models more efficient.

### 4.1 TRANSFORMER NETWORKS

Transformer networks were introduced by Vaswani et al. (2017a) in the context of machine translation. These models are made of a sequence of identical blocks, each comprising two sublayers. The first sublayer, called `SelfAttention`, is based on the attention mechanism from Bahdanau et al. (2014). More specifically, given a sequence of $T$ hidden states of dimension $d$, represented by the matrix $\mathbf{X} \in \mathbb{R}^{d \times T}$, the output of this layer is given by

$$\mathbf{Z} = \mathbf{W}_v \mathbf{X} \mathtt{Softmax} \left( \mathbf{X}^T \mathbf{W}_k^T \mathbf{W}_q \mathbf{X} \right),$$

where $\mathbf{W}_v, \mathbf{W}_k$ and $\mathbf{W}_q$ are the parameters of the sublayer used to compute the values, keys and queries of the attention mechanism. Multiple self-attention mechanisms are usually applied in parallel, leading to the multi-head self-attention sublayer.

The second sublayer, called `FFN`, is a fully connected feed forward network with RELU activation, which is applied independently at each time step. Each sublayer is followed by a skip connection and a layer norm operator, to finally obtain:

$$\mathbf{Z} = \mathtt{LayerNorm}(\mathtt{SelfAttention}(\mathbf{H}) + \mathbf{H}),$$

and

$$\mathbf{Y} = \mathtt{LayerNorm}(\mathtt{FFN}(\mathbf{Z}) + \mathbf{Z}).$$

Following Dai et al. (2019), we use relative positional encodings at each layer, as well as a cache mechanism, to make the processing of long sequences more efficient. In the following, we will refer to this model as Transformer-XL. We refer the reader to Vaswani et al. (2017a) and Dai et al. (2019) for a more detailed introduction to these models.

### 4.2 MODIFICATIONS TO THE TRANSFORMER-XL MODEL

We propose several modifications to the Transformer-XL model that we found to improve the compression rate empirically. First, we follow Devlin et al. (2018) and use the GELU activation of Hendrycks & Gimpel (2016) instead of ReLUs. Second, as opposed to Transformer-XL (Dai et al., 2019), we do not use Adaptive Inputs (Baevski & Auli, 2018) to tie the weights of embeddings and the classifier. Finally, we use different attention span size across all layers as was suggested by Sukhbaatar et al. (2019a). In particular, an observation from this work is that attention span can be short in the first half of the network while it has to be longer in the last layers. We exploit this observation by restricting the attention span of the first layers of the Transformer and is extended linearly with the depth. More specifically, we restrict the attention size of the first 6 layers to be equal to $6 \times k$ for the $k$-th layer.

### 4.3 ADDING $n$-GRAMS AS INPUT

We also propose to enrich the input of the transformer with $n$-gram information. More specifically, at time step $t$, the input of the model is all the $n$-grams for $n$ between $1$ and $N$ of the form

| Method | BPC |
|---|---|
| *Standard text compression* | |
| `gzip` -9 | 2.92 |
| `xz` -9 | 1.99 |
| paq8hp12any | 1.31 |
| CMIX | **1.20** |
| *Neural Network based compression* | |
| `lstm-compress` | 1.65 |
| Transformer Bellard (2019) | 1.46 |
| LSTM Bellard (2019) | 1.35 |
| ours without revisit | 1.33 |
| ours with revisit | **1.20** |

Table 1: Compression rate on `enwik8` for different methods

$x_{t-n+1}, ..., x_t$. Concretely, we have an embedding vector for each $n$-gram, and for a given time step, we average the vectors corresponding to the $n$-grams of different lengths. For large $n$, there are many rare $n$-grams, leading to an important increase of parameters and memory consumption. A potential solution would be to only keep the most frequent ones, but this would require computing a dictionary of $n$-grams and add it to the archive. Instead, we propose to hash the $n$-grams into a fixed number of bins, allowing to constrain the memory requirement, without having to increase the size of the archive.

## 4.4 TRAINING WITH OR WITHOUT REVISITS

During the compression phase (resp. decompression), once a token has been encoded (resp. decoded), it can be used to update the probabilistic model. This means that compression using statistical models can be framed as an online learning problem. Traditionally, previously encoded tokens are used only once to update the probabilistic model. However, nothing prevents re-using the tokens to compute multiple updates of the model, besides memory consumption. Indeed, to be able to do so, one needs to store the data being compressed to be able to revisit it. In the following we propose to explore this strategy to learn better probabilistic models based on transformer networks. The main motivation is that large overparametrized neural networks usually needs many passes over the data to be trained. In practice, every $F$ compressed tokens, we re-use the last $M$ compressed tokens to further train the network. This strategy will improve the quality of the probabilistic model, leading to better predictions and compression rate of the data not yet processed. This phase is determined by the frequency of revisit $F$ and the number of tokens revisited $M$.

## 5 EXPERIMENTS

In this section, we investigate the influence of different parameters on the performance of our approach. We also provide numerical evidence that a purely neural based approach can be competitive with state-of-the-art methods in terms of compression ratio.

## 5.1 IMPLEMENTATION DETAILS

**Dataset.** The standard dataset for text compression is `enwik8`, notably used for the Hutter Prize. It is composed of 100MB of English Wikipedia data. In the compression setting, we report the bit per character (BPC) obtained the first time we visit each token of the full dataset. This differs from language modeling where we report the BPC on withheld data.

**Preprocessing.** We use the CMIX preprocessor where all uppercase letters are converted to lowercase ones preceded by an escape code. This reduces the size of the vocabulary from 205 characters to 165 characters, while increasing the number of symbols by approximately 3% on `enwik8`.

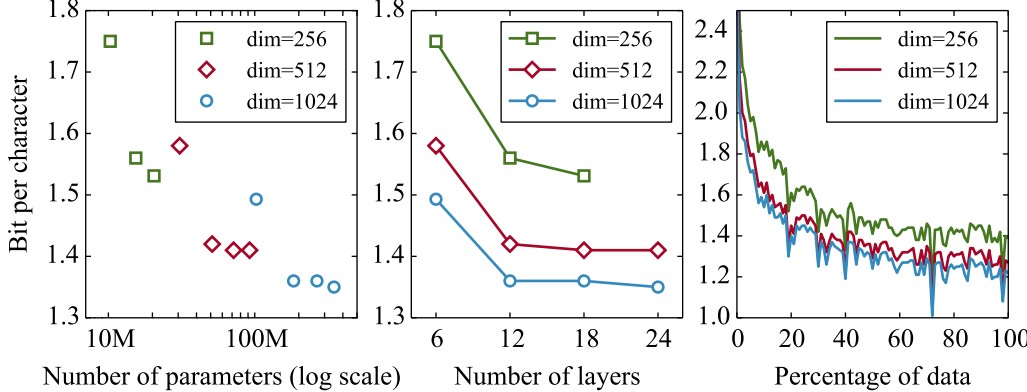

Figure 1: Impact of the model size on compression rate. We report the compression rate in BPC for transformer networks of various sizes. In particular, we vary the number of layers (6, 12, 18, 24) as well as the dimension of each layer (256, 512, 1024). Left: we observe that on the full `enwik8` dataset, larger models obtain better compression rate. Middle: using models deeper than 12 layers does not seem to improve the compression. Right: Instantaneous compression rate (averaged over the last 1 million characters) for 12 layers models. Larger models obtain better performance, even at the beginning of compression.

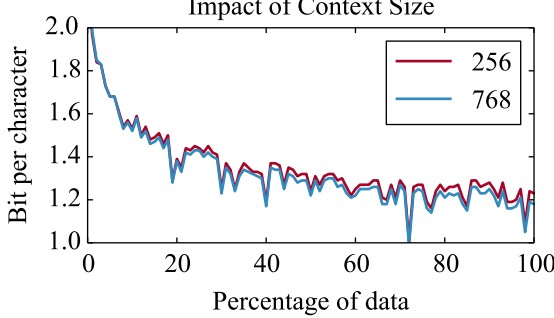

Figure 2: Instantaneous compression rate (averaged over the last 1 million characters) for models with different context sizes.

| Context size | BPC |
|---|---|
| 256 | 1.36 |
| 512 | 1.34 |
| 768 | **1.33** |

Table 2: Compression rate on the full `enwik8` data for models with different context sizes.

**Transformer.** We consider Transformers with either 12, 18 or 24 layers, a hidden size $d$ of either 256, 512 or 1024, and a number of heads between 4, 8 and 16. The dimension of the feedforward $d_f$ is $4d$. We consider several size for the span, i.e., 256, 512 and 768 , but in practice longer spans work better. The optimizer used to train the Transformer impacts greatly the compression ratio. Our best results were obtained with Adam optimizer Kingma & Ba (2015) with the correction introduced in Reddi et al. (2018) and its variant Adamw Loshchilov & Hutter (2017). The impact of the different hyperparameters involved in these optimizer is investigated in the following subsections.

## 5.2 RESULTS

In table 1 we summarize the performance obtained with transformer networks in this report. the fact that other approaches require the transmission of a dictionary to the decoder makes the comparison difficult: in most cases it is not accounted in the performance although it can represent up to 400kb of uncompressed data. in order to obtain a comparison that makes sense, we take into account the size of the dictionary used in the different methods after compression with `gzip -9`. the code used to train the different networks also needs to be transmitted to the decoder, but its size is negligible.

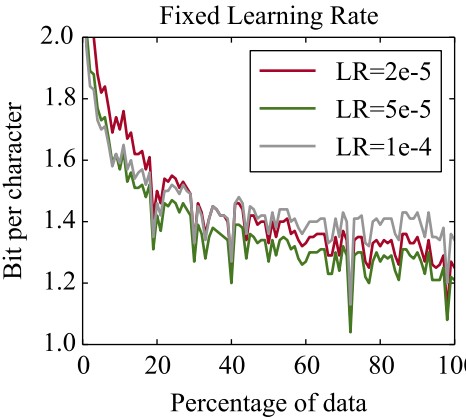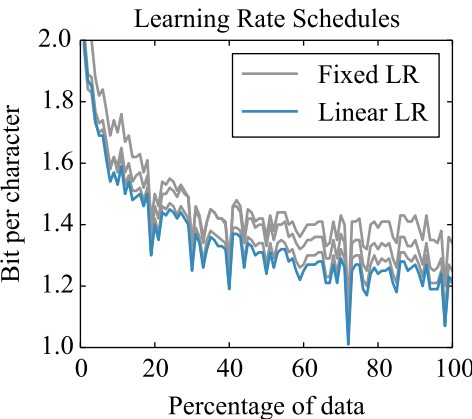

Figure 3: Instantaneous compression rate (averaged over the last 1 million characters) for different fixed learning rates (left) and linear decay schedule (right). We observe that higher learning rates get better compression rate at the beginning of the compression phase, but lower learning rates tend to obtain better results at the end. A linear decay schedule allows to match the best performing fixed learning rate at most steps of the compression.

## 5.3 IMPACT OF THE NUMBER OF PARAMETERS

First, we investigate the influence of the model size, as well as its depth and layer dimension, on the compression rate. In many previous work, it has been observed that large overparametrized models lead to state-of-the-art performance. For example, Vaswani et al. (2017b) notice that large transformers with up to 160M parameters obtain better results than smaller networks. Similarly, for language modeling, Dai et al. (2019); Sukhbaatar et al. (2019a); Radford et al. (2019) present better results with larger networks. However, our online learning setting is different from these works, and a natural question is whether this observation is also true for compression. More specifically, it could be possible that smaller networks would learn faster, as less parameters needs to be trained.

In Figure 1, we report the compression rate on `enwik8` obtained by transformers of various sizes. The rest of the parameters, such as the optimization parameters, are identical for all runs and are close to the best we have found. First, we observe that, as in the supervised learning setting, larger models do obtain better performance. In particular, our best model was also the largest one, containing almost 350M parameters.

More surprisingly, larger models also obtained better compression ratio, even at the very beginning of the compression phase. In Figure 1, we also report the BPC averaged over the last 1M characters processed by the model (instantaneous compression rate). We compare three networks made of 12 layers: a small network with 4 heads, a hidden state of dimension $d = 256$, a medium network, with 8 heads and a hidden state of dimension 512 and finally, a large network, with 16 heads and a hidden state of dimension 1024. Even after processing only one percent of the data, the large model have a better compression rate than the small and medium models.

Finally, we study the influence of the depth of the network on the compression rate. In Figure 1, we report results obtained by networks of various depths, all the other dimensions being equal. Here, we observe that using models deeper than 12 layers do not really improve the compression rate. In particular, as opposed to other sequence modeling tasks, it seems more efficient to consider wider networks than deeper ones, for a constant number of parameters.

## 5.4 IMPACT OF THE CONTEXT SIZE

The size of the context in transformer networks determines the maximum number of previous hidden states the model can use to predict the next symbol. In Table 2, we compare the compression rate obtained by using different context sizes, from 256 to 768. We show that larger context size results in better performance overall. A larger memory provides to the network more information, but it

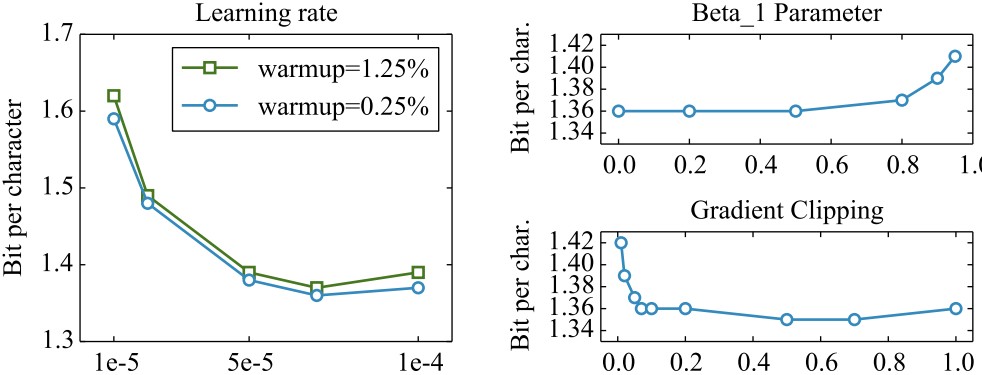

Figure 4: Impact of different optimization parameters. Left: we report the BPC over the full `enwik8` data for different learning rate with linear decay and number of warmup steps. Top right: small $\beta_1$ values for ADAM variants lead to better compression rate on the full `enwik8`. Bottom right: as for supervised learning, gradient clipping is helpful in the compression setup.

could also be more difficult to exploit: the most useful information to predict the next character is usually located in the few preceding characters. In particular, we observe that at the beginning of the compression phase, models with shorter contexts tend to perform better. To illustrate this, we report in Figure 2 the instantaneous BPC for models with context sizes 256 and 768. Finally, it is important to note that the computation time and the memory scale linearly with the size of the context.

## 5.5 IMPACT OF THE LEARNING RATE SCHEDULE

In general, the learning rate is a parameter having an important influence on the convergence speed of neural networks. In Figure 3, we report the compression rate, averaged over the last one million characters, obtained by using different fixed learning rate with the AMSGrad algorithm. As expected, a higher learning rate leads to a better compression rate at the beginning of the run, but a smaller learning seems important to obtain good performance at the end of the compression phase. In order to have the best of both worlds, we use a linear decay schedule, such that the learning rate is equal to zero at the end of the compression phase. In Figure 3, we observe that this scheme outperforms fixed learning rate schedules over most of the compression. Overall, the best fixed learning rate leads to a compression rate of 1.38 BPC, while linear decay get to 1.36 BPC.

When training transformers, it is standard practice to use a warmup phase at the beginning of learning, during which the learning rate increases from zero to its peak value (Vaswani et al., 2017b). We follow this practice, meaning that our linear learning schedule is determined by two parameters: the peak learning rate and the length of the warump phase. We report the influence of these two parameters on the overall compression rate in Figure 4. It appears that a longer warmup phase is detrimental to the performance of transformer based data compression.

## 5.6 OTHER OPTIMIZER PARAMETERS

Finally we found that the compression ratio is sensitive to the optimizer parameters. In this section, we evaluate the impact of the following parameters of the Adam optimizer: $\beta_1$, $\beta_2$ and $\varepsilon$. We report the BPC over the full `enwik8` data for different values of $\beta_1$ in Figure 4. The fact that small $\beta_1$s give the best results is surprising, as usually, values close to 1.0 are used. For example, the default $\beta_1$ value in PyTorch is 0.9, and gives a compression rate of 1.39 BPC, compared to 1.36 for the optimal value of $\beta_1$. On the other hand, we found the $\beta_2$ and $\varepsilon$ parameters to have little influence on the compression rate, and standard values to give the best results. Finally, Figure 4 shows that gradient clipping is helpful, but the performance are not very sensitive to its value.

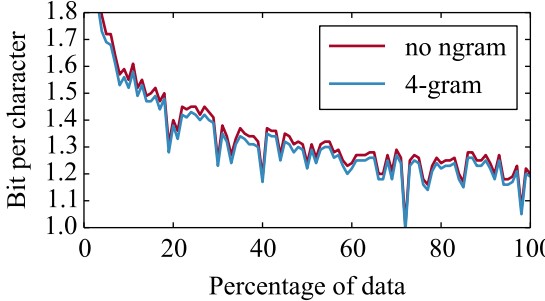

| $k_{\max}$ | Number of vectors | BPC |
|---|---|---|
| 4 | 20k | 1.337 |
| 4 | 50k | 1.337 |
| 3 | 80k | 1.334 |
| 4 | 80k | 1.335 |
| No $n$-gram information | | 1.360 |

Figure 5: Instantaneous compression rate (averaged over the last 1 million characters) for models with and without $n$-grams. We observe that adding $n$-gram information lead to a better compression rate for the whole compression phase.

Figure 6: Compression ratio depending on $k_{\max}$ and the number of vectors used to represent the $n$-grams

### 5.7 IMPACT OF PREPROCESSING AND $n$-GRAMS

We only apply a minimal preprocessing step to the data, consisting in replacing uppercase characters by lowercase ones, with an additional escape code. This simple step, also used by other compression algorithms such as CMIX, leads to an improvement of 0.01 BPC. We also considered adding $n$-gram information to the input of the transformer model. At each time step $t$, the input corresponds to the current character, as well as the $n$-grams ending at position $t$. To avoid storing a dictionary of $n$-grams in the archive, we instead hash the $n$-grams into one a fixed number of bins, each bin being associated to a learned embedding vector. We report the effect of adding $n$-grams in Table 6 and Figure 5. We show that adding this information lead to an improvement of 0.025 BPC, and this improvement can still be observed at the end of the compression phase. By combining the optimal parameters derived from the different experiments reported in this section, we can compress `enwik8` using 1.33 bits per character. While this is far from the performance of CMIX, it improves previous results obtained with transformer networks (Bellard, 2019).

### 5.8 IMPACT OF REVISITING DATA

Finally, we study the impact of revisiting data already compressed, to improve the quality of the model and its predictions on data to be processed. Overall, this is a critical strategy that allows to improve the compression rate from 1.33 BPC to 1.20 BPC, for the full `enwik8` dataset. For this, every 512k compressed characters, we re-use the last 12.8M characters with batch size of 32 to further train the network. During the revisit phase, we use dropout regularization, with a rate of $p = 0.25$, so that the model does not overfit the past data. Introducing the use of dropout during the revisit improves the compression rate by approximately 0.05 BPC.

## 6 CONCLUSION

In this paper, we explore the application of transformer networks to the problem of lossless data compression, which is closely related to the online learning of sequence models. One challenge of using large neural networks to this task is the fact that many passes over the data are usually required to train them. We thus propose to periodically revisit tokens that were already compressed to further train the model, and improve its predictions. We also perform a thorough study of the impact of various hyper-parameters, such as the model architecture, on the compression rate. Based on this study, we show that a transformer based approach obtains state-of-the-art results on the competitive `enwik8` benchmark, without using hand crafted features.

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
