# OpenReview forum: "Lossless Data Compression with Transformer"
_ICLR.cc/2020/Conference — Reject_

### Official Review · AnonReviewer3 · 2019-10-23
**Official Blind Review #3**

**Rating:** 3

**Review:**

Summary:
The paper investigates using the transformer architecture for neural network-based lossless compression of text. The resulting model, obtained through a thorough investigation of the architecture hyper-parameters are on par with standard SOTA compression. The paper is well-written. In particular the authors have done a great job reviewing existing compression literature and positioning their method within the space of prior work.

Recommendation: Weak Reject
While the paper considers an interesting application of the Transformer architecture, and is well-written, it is of limited novelty. Specifically, the bulk of the paper is concerned with describing experimental results of a thorough (but standard) hyper-parameter search - considering things like Transformer context size, learning rate (schedule), number of layers and key, value, query dimensionality; and does not offer any new architectural modifications / insights.

Furthermore, only a single dataset - enwik8 - is considered in the experimental validation and little attention is given to the description of the dataset split and any distribution differences between splits. Taken together, the existing experimental setup potentially creates an unfair advantage for the neural network-based methods - while the standard methods can be expected to perform similarly across a wide range of datasets / texts, the neural-network based methods have been trained and tested on very similar data and could be expected to perform well on these data, but not in case of a distributional shift (e.g. compressing legal texts instead of Wikipedia). The paper does not answer the question of whether or not this is true.

Furthermore, similar to autoregressive models, transformers are known to be slow at inference time. I expect this to lead to very slow decoding. Therefore, methods in table 1 should be compared in compression/decompression time to give a better overview of the practical impact of this work.

Taken together, in its current form the paper may be better suited for a workshop publication rather than a full conference paper.

Major comments:
1. For reasons mentioned above, the paper should include additional experimental evaluation. In particular, it should consider the effect of training the model on one dataset, but evaluating it on another dataset; and discuss how differences in performance (if any) compare to standard methods.
2. Compression/decompression times of the proposed method should be compared against the other compression methods in table 1. I expect the proposed transformer to be slow at decompressing.
3. The paper does not contain the loss that the transformer model was used to optimize. I assume that it is the softmax cross entropy, but this is worth mentioning explicitly. It would also be worthwhile to explain the training procedure - for how many epochs was the model trained (see also next question), what was the dataset size?
4. Description of the “training with revisits” is not very clear. My understanding is that it resembles a pass through the data, where some of it is considered again at specific intervals. My first assessment is that this should not be necessary - the data should already be considered multiple times during the training process.
a) The authors should provide a more detailed description of the training-with-revisits procedure, contrasting it specifically with a procedure where revisits are not done (i.e. normal training).
b) If the goal of the revisits training is to observe some training examples more than once, then it would be very interesting if simply training for a longer time (several epochs == passes through the data) has a similar effect.
c) Is there any motivation for the choice of the revisits hyper-parameters F and M? Was a different batch size used during the revisits training? Is the learning rate evolved during the revisits training phase or is it still decayed?

Minor comments:
1. There is some prior work on using Neural Networks for lossless image compression (e.g. [1], [2]. [3] that achieves SOTA compression ratios compared to standard methods. It may be interesting for the readers to mention these results. In particular the authors’ statement that “[...] purely neural network based models are still far from state of the art [...]” may give the wrong impression to the readers.
2. The authors mention that they “[...] propose several improvements to its (the Transformer) architecture and training to accelerate and stabilize [...] training”. In my view, the experiments described in the paper resemble a hyper-parameter search more than architectural improvements. The authors may want to clarify in the text which specific improvements they refer to.
3. Page 1, last paragraph: “[...] of all the important component [...]” -> “[...] of all the important components [...]”
4. Page 3: “[...] attention span size across all layers as it suggested [...]” -> “[...] attention span size across all layers as was suggested [...]”
5. Page 3: Missing references.
6. Page 3: Use of small n and capital N when talking about n-grams. Should be made consistent.
7. Page 8 (Conclusion): “wihtout” -> “without”


[1] F. H. Kingma, P. Abbeel, and J. Ho. Bit-Swap: recursive bits-back coding for lossless compression with hierarchical latent variables. In International Conference on Machine Learning (ICML), 2019.
[2] Emiel Hoogeboom, Jorn W. T. Peters, Rianne van den Berg, and Max Welling. Integer Discrete Flows and Lossless Compression. arXiv e-prints, 2019.
[3] Jonathan Ho, Evan Lohn, and Pieter Abbeel. Compression with Flows via Local Bits-Back Coding. arXiv e-prints, 2019.


**Experience Assessment:**

I have published one or two papers in this area.

**Review Assessment: Checking Correctness Of Derivations And Theory:**

N/A

**Review Assessment: Checking Correctness Of Experiments:**

I carefully checked the experiments.

**Review Assessment: Thoroughness In Paper Reading:**

I read the paper at least twice and used my best judgement in assessing the paper.

---

> ### Author Response · Authors · 2019-11-15
> **Response to Review #3**
>
> We thank the reviewer for their feedback.
>
> 1. We will add experiments on the silesia benchmark (http://mattmahoney.net/dc/silesia.html). Overall, our method compress the whole corpus in 33.4MB, compared to 33.3MB for cmix (we report bpc for individual files below).
>
>          	| paq8 | cmix |	ours
> dickens  | 1.58 | 1.49 | 1.59
> mozilla	| 1.61 | 1.49 | 1.41
> mr    	| 1.68 | 1.62 | 1.48
> nci   	| 0.23 | 0.20 | 0.20
> ooffice	| 1.87 | 1.74 | 2.25
> osdb  	| 1.65 | 1.60 | 1.58
> reymont	| 0.98 | 0.93 | 0.94
> samba 	| 1.02 | 0.96 | 1.14
> sao   	| 4.16 | 4.16 | 4.18
> webster	| 1.00 | 0.90 | 0.84
> x-ray 	| 3.41 | 3.37 | 3.33
> xml   	| 0.41 | 0.37 | 0.54
>
> Similarly to cmix and PAQ8, our method don’t pretrain the language model on a dataset different from the data to compress. The neural network is randomly initialized at the beginning of the compression and decompression phase with the same seed and is trained during both phases. The only constraint is that a given character has first to be compressed before being used to update the model. We have considered initializing the neural network with a pre-trained model. However in order to decompress the archive, the decoder needs to have access to the pretrained model used by the encoder. Thus, the size of the pre-trained model has to be added to the size of the archive. As a result, we didn’t manage to improve performances by using a pre-trained model.
>
> 2. There is an inherent tradeoff between compression ratio and compression/decompression speed. In this paper we focus on the compression ratio, and at the expense of speed we match the performance of CMIX on enwik8. We will add runtime in the paper.
>
> 3. Indeed, the model is trained using the softmax cross entropy. Without revisit, the weights of the network is updated every 256 characters. The gradient is obtained by backpropagating the error of the prediction associated with the 256 characters that have just been compressed. Thus during compression and decompression the network is trained for one epoch. With revisits, the number of times a given character is used to train the Transformer is increased. Enwik8 is composed of 100MB of wikipedia data in XML format.
>
> 4. a) A revisit is a partial pass on data that have already been compressed. Revisits are performed during compression and decompression at fixed intervals to further train the network. All weight updates performed during compression have to be performed identically during decompression in order to losslessly decompressed data. Thus increasing the number of revisits makes compression and decompression slower. In particular since it is necessary to compute a forward pass to compress the data, the cost of backpropagating the error of the first prediction is amortized.
> b) Since we made the assumption (also made in PAQ8 and CMIX) that pretrained models used for compression have to be included in the archive in order to be accessed by the decoder, revisits is just the way to use data several times to learn the parameters of the model.
> c) During a revisit the learning rate is fixed, but it is linearly decreased over the compression phase. If the frequency of revisit F is too low, a higher revisit frequency could improve the compression ratio, especially at the beginning of the compression. We have observed that at some point increasing the frequency of revisit can be detrimental to the compression ratio. This can be explained by the fact that the network tends to forget the current context. Increasing the number of characters considered at each revisits improve the compression ratio but is detrimental to the compression/decompression speed.
>
> We will address the minor comments in the paper, and will add the missing references to the related work.

---

### Official Review · AnonReviewer1 · 2019-10-24
**Official Blind Review #1**

**Rating:** 1

**Review:**

This paper provides a method for lossless compression of text. It's heavily inspired by the language modelling methods that have been developed for the purposes of predicting the next character/word in a sentence, and it uses this idea as its backbone. The only difference is that the results are presented in the compression setting.

I think we should reject this paper due to the following reasons:
- I don't see enough of a difference between this and previous work
- the results are nowhere near SoTA for compression, despite the method being sold to this community
- there are other papers that do lossless neural compression that could have been used to make a comparison rather than making no comparison at all. For example, "Practical Full Resolution Learned Lossless Image Compression" (CVPR 2019) provides a framework for image rather than text, but that could be adapted to this field without any major changes (predict convolutionally characters, rather than RGB values).
- there's no comparison even with BERT (how well it do to predict the next character vs. this)...
- no runtime numbers
- no reproducibility discussion (i.e., how can I guarantee that my decoder can get exactly the same numbers as my encoder so that I can decompress on a different machine)
- no discussion about whether files were created/decompressed (this is ABSOLUTELY CRUCIAL for compression papers to discuss)

Overall, I am not excited about this paper, and unless the authors put a lot more into it, there's just not enough novelty to justify a publication at ICLR.

**Experience Assessment:**

I have published in this field for several years.

**Review Assessment: Checking Correctness Of Derivations And Theory:**

N/A

**Review Assessment: Checking Correctness Of Experiments:**

I assessed the sensibility of the experiments.

**Review Assessment: Thoroughness In Paper Reading:**

I read the paper at least twice and used my best judgement in assessing the paper.

---

> ### Author Response · Authors · 2019-11-15
> **Response to Review #1**
>
> We thank the reviewer for their feedback.
>
> In this paper, we show that a method purely based on neural networks, without hand designed features, can obtain SoTA results compression results on benchmarks such as enwik8. Existing work showed a significant gap between methods purely based on neural networks, compared to methods such as PAQ8 or cmix (see https://bellard.org/nncp/nncp.pdf). Also note that for each application, there is a tradeoff between compression ratio and compression/decompression speed. Our focus in this paper is to obtain the best possible compression rate, at the expense of compression speed.
>
> Regarding SoTA results for compression, cmix and PAQ8 variants are the methods obtaining the best compression rates, according to https://cs.fit.edu/~mmahoney/compression/text.html,  http://mattmahoney.net/dc/silesia.html or http://qlic.altervista.org/LPCB.html. Could the reviewer indicates methods outperforming these approaches that we might have missed? Note that results reported on enwik8 in the traditional language modeling setting (i.e. training on 90% of data, validating and testing on the rest) are not comparable to the compression setting we study.
>
> We will add a discussion of "Practical Full Resolution Learned Lossless Image Compression" [1] as well as the paper mentioned by reviewer 3 in the related work. The method proposed for image compression in [1] combines arithmetic coding with a neural network. As opposed to our work, the approach is designed to enable practical compression and decompression speed with compression ratio comparable with standard methods (while we focus on compression rate only). Another difference with our work is the evaluation setting: in [1] it is assumed that both the encoder and the decoder have access to the pre-trained network for free. As PAQ8 and cmix, we do not use a pre-trained network, it has to be included in the archive in order to be accessed by the decompressor. While it is true that sending a network to the decoder can be amortized over the decompression of large amount of data, the size of the L3C network archive is 35MB and not negligible compared to the size of the compressed archive of enwik8, around 15MB. It should also be noted that PAQ8 and CMIX achieve better compression ratio than other compression algorithms on a dataset composed of large images, at the expense of compression and decompression speed (e.g. see http://qlic.altervista.org/LPCB.html).
>
> BERT fundamentally differs from our setting in several aspects. First, BERT is not trained as a language model to predict the next character given the preceding characters, but as a denoising auto-encoder. As such, BERT is not a generative model of sequence, and it is not straightforward to apply it to data compression. Moreover, standard BERT models are several 100s of MB in size, which would need to be included in the archive to allow the decompression. As such, it does not make using BERT practical for lossless data compression.
>
> We have not yet implemented an end-to-end framework for compression and decompression. The decoder and encoder can get the same numbers provided that they have access to the same random number generator and the same seed. Finally, we will release our code with the paper to allow reproducibility.

---

### Official Review · AnonReviewer2 · 2019-10-25
**Official Blind Review #2**

**Rating:** 3

**Review:**



This paper explores the effectiveness of the Transformer architecture to the lossless data compression problem.
It also proposes a method to periodically revisit tokens that were already compressed for adopting the task setting of data compression, which is essentially online learning of sequence models.

The authors conduct their experiments on the enwik8 benchmark.
They show that the Transformer architecture obtains state-of-the-art results.

This paper is basically easy to follow, but several typos and statements that should be improved.
The problem setting to tackle is interesting.
However, applying a deep neural network approach to data compression problem has already been discussed in several previous studies.
Therefore, the novelty of this paper is somewhat limited.


My main concern of this paper is that the proposed method was only evaluated on a single benchmark data.
I believe that it is a bit weak to support the effectiveness of the proposed method.
The authors should evaluate their method on several benchmark datasets that have different aspects, such as settings with easy and hard to compress.


Minor comment:
In Section 4.2, there is a missing citation.
... we do not use Adaptive Inputs (Baevski & Auli, 2018; ?) ...
Please check and fix it.



**Experience Assessment:**

I have read many papers in this area.

**Review Assessment: Checking Correctness Of Derivations And Theory:**

N/A

**Review Assessment: Checking Correctness Of Experiments:**

I carefully checked the experiments.

**Review Assessment: Thoroughness In Paper Reading:**

I read the paper thoroughly.

---

> ### Author Response · Authors · 2019-11-15
> **Response to Review #2**
>
> We thank the reviewer for their feedback.
>
> We will update the paper to address the minor comments. Moreover, we evaluated our method on the additional Silesia benchmark (http://mattmahoney.net/dc/silesia.html), which includes files of different types (such as text, UNIX or Windows executables, databases, pdf etc). We report compression rates for our method, as well as PAQ8L and cmix v8 below, and will add it to the paper. Overall, our method compress the whole corpus in 33.4MB, compared to 33.3MB for cmix. We report bpc for individual files below.
>
>          	| paq8 | cmix |	ours
> dickens  | 1.58 | 1.49 | 1.59
> mozilla	| 1.61 | 1.49 | 1.41
> mr    	| 1.68 | 1.62 | 1.48
> nci   	| 0.23 | 0.20 | 0.20
> ooffice	| 1.87 | 1.74 | 2.25
> osdb  	| 1.65 | 1.60 | 1.58
> reymont	| 0.98 | 0.93 | 0.94
> samba 	| 1.02 | 0.96 | 1.14
> sao   	| 4.16 | 4.16 | 4.18
> webster	| 1.00 | 0.90 | 0.84
> x-ray 	| 3.41 | 3.37 | 3.33
> xml   	| 0.41 | 0.37 | 0.54

---

### Decision · Program_Chairs · 2019-12-19

**Decision:**

Reject

**Comment:**

The paper proposes to use transformers to do lossless data compression. The idea is simple and straightforward (with adding n-gram inputs). The initial submission considered one dataset, a new dataset was added in the rebuttal. Still, there is no runtime in the experiments (and Transformers can take a lot of time to train). Since this is more an experimental paper, this is crucial (and the improvements reports are very small and it is difficult to judge if there are significant).
Overall, there was a positive discussion between the authors and the reviewers. The reviewers commented that concerns have been addressed, but did not change the evaluation which is  unanimous reject.